Data science competition for cross-site individual tree species identification from airborne remote sensing data

http://orcid.org/0000-0003-3805-4242 Graves Sarah J. 1 sjgraves142@gmail.com
Marconi Sergio 2
http://orcid.org/0000-0003-4505-657X Stewart Dylan 3
Harmon Ira 4
Weinstein Ben 2
Kanazawa Yuzi 5
http://orcid.org/0000-0002-2085-1449 Scholl Victoria M. 6 7
http://orcid.org/0000-0002-7745-9990 Joseph Maxwell B. 6
McGlinchy Joseph 6
http://orcid.org/0000-0002-2239-3007 Browne Luke 8
http://orcid.org/0000-0001-9926-1929 Sullivan Megan K. 8
Estrada-Villegas Sergio 8
Wang Daisy Zhe 4
Singh Aditya 9
Bohlman Stephanie 10
http://orcid.org/0000-0002-4847-7604 Zare Alina 3 11 12
http://orcid.org/0000-0001-6728-7745 White Ethan P. 2 11 12
1 Nelson Institute for Environmental Studies, University of Wisconsin-Madison , Madison, Wisconsin , United States
2 Department of Wildlife Ecology and Conservation, University of Florida , Gainesville, Florida , United States
3 Department of Electrical and Computer Engineering, University of Florida , Gainesville, Florida , United States
4 Department of Computer and Information Sciences and Engineering, University of Florida , Gainesville, Florida , United States
5 Artificial Intelligence Laboratory, Fujitsu Laboratories Ltd. , Kawasaki, Kanagawa , Japan
6 Earth Lab, Cooperative Institute for Research in Environmental Sciences (CIRES), University of Colorado at Boulder , Boulder, Colorado , United States
7 Department of Geography, University of Colorado at Boulder , Boulder, Colorado , United States
8 Yale School of the Environment, Yale University , New Haven, Connecticut , United States
9 Department of Agricultural & Biological Engineering, University of Florida , Gainesville, Florida , United States
10 School of Forest, Fisheries, and Geomatics Sciences, University of Florida , Gainesville, Florida , United States
11 Informatics Institute, University of Florida , Gainesville, Florida , United States
12 Biodiversity Institute, University of Florida , Gainesville, Florida , United States
Oehlmann Jörg
Electronic publication date: 2023 Dec 21
Publication date: 2023
Volume: 11
Electronic Location ID: e16578
Received 2023 Apr 25; Accepted 2023 Nov 13
Copyright: © 2023 Graves et al.
Copyright year: 2023
Copyright holder: Graves et al.
License: This is an open access article distributed under the terms of the Creative Commons Attribution License, which permits unrestricted use, distribution, reproduction and adaptation in any medium and for any purpose provided that it is properly attributed. For attribution, the original author(s), title, publication source (PeerJ) and either DOI or URL of the article must be cited.
License URL: https://creativecommons.org/licenses/by/4.0/

Keywords: Airborne remote sensing, Species classification, National ecological observatory network, Data science competition

Funding: National Science Foundation Gordon and Betty Moore Foundation’s Data-Driven Discovery Initiative GBMF4563 & 1926542 NSF Dimension of Biodiversity Program Grant DEB-1442280 USDA/NIFA McIntire-Stennis Program FLA-FOR-005470 University of Florida Biodiversity Institute (UFBI) Informatics Institute (UFII) Graduate Fellowship The National Ecological Observatory Network is a program sponsored by the National Science Foundation and operated under cooperative agreement by Battelle. This material is based in part upon work supported by the National Science Foundation through the NEON Program. This work was supported by the Gordon and Betty Moore Foundation’s Data-Driven Discovery Initiative through grant GBMF4563 to Ethan P White; by the National Science Foundation through grant 1926542 to Ethan P White, Stephanie Bohlman, Alina Zare, and Daisy Z Wang; by the NSF Dimension of Biodiversity program grant (DEB-1442280) and USDA/NIFA McIntire-Stennis program (FLA-FOR-005470) to Stephanie Bohlman; by the University of Florida Biodiversity Institute (UFBI) and Informatics Institute (UFII) Graduate Fellowship to Sergio Marconi. The funders had no role in study design, data collection and analysis, decision to publish, or preparation of the manuscript.

==============================
Data on individual tree crowns from remote sensing have the potential to advance forest ecology by providing information about forest composition and structure with a continuous spatial coverage over large spatial extents. Classifying individual trees to their taxonomic species over large regions from remote sensing data is challenging. Methods to classify individual species are often accurate for common species, but perform poorly for less common species and when applied to new sites. We ran a data science competition to help identify effective methods for the task of classification of individual crowns to species identity. The competition included data from three sites to assess each methods’ ability to generalize patterns across two sites simultaneously and apply methods to an untrained site. Three different metrics were used to assess and compare model performance. Six teams participated, representing four countries and nine individuals. The highest performing method from a previous competition in 2017 was applied and used as a baseline to understand advancements and changes in successful methods. The best species classification method was based on a two-stage fully connected neural network that significantly outperformed the baseline random forest and gradient boosting ensemble methods. All methods generalized well by showing relatively strong performance on the trained sites (accuracy = 0.46–0.55, macro F1 = 0.09–0.32, cross entropy loss = 2.4–9.2), but generally failed to transfer effectively to the untrained site (accuracy = 0.07–0.32, macro F1 = 0.02–0.18, cross entropy loss = 2.8–16.3). Classification performance was influenced by the number of samples with species labels available for training, with most methods predicting common species at the training sites well (maximum F1 score of 0.86) relative to the uncommon species where none were predicted. Classification errors were most common between species in the same genus and different species that occur in the same habitat. Most methods performed better than the baseline in detecting if a species was not in the training data by predicting an untrained mixed-species class, especially in the untrained site. This work has highlighted that data science competitions can encourage advancement of methods, particularly by bringing in new people from outside the focal discipline, and by providing an open dataset and evaluation criteria from which participants can learn.

Introduction

High resolution remote sensing imagery provides critical information about the presence and types of organisms within and among ecosystems at scales beyond those observable using field techniques. Inventory data from remote sensing, such as the location, size, and species identity of individual trees is useful for ecological studies and the management of forests (White et al., 2016), including studies of population dynamics (Clark et al., 2004; Kellner & Hubbell, 2018), vegetation phenology (Wu et al., 2016; Park et al., 2019), biomass and carbon (Duncanson et al., 2015; Jucker et al., 2016), foliar properties (Zheng et al., 2021; Marconi et al., 2021), and species composition and biodiversity (Baldeck et al., 2014; Rocchini et al., 2016; Baena et al., 2017). While it is often useful to gather information from remote sensing at the stand or forest level, critical processes for ecology, ecosystem services and wood production, such as growth and mortality, occur at the individual scale.

Species identity at the individual scale is crucial for models of biodiversity, and plays a significant role in parameterizing models for ecosystem services, habitat modeling, and forestry (Duncanson et al., 2015; Barber et al., 2022). Species classification models have been a long-standing challenge in remote sensing of forests, with complexity especially in dense forests, due to weakly defined edges among trees, large intra-class variance in tree representation, and high local diversity within a forest. Early work used spectral features, texture-based features, random forests and support-vector-machines, band ratios within the visible and NIR, and other crafted-features (Fassnacht et al., 2016; Ballanti et al., 2016; Shi et al., 2018; Modzelewska, Fassnacht & Stereńczak, 2020). These initial models often focused on higher-order taxonomic labels, such as ‘Birch’ vs. ‘Pine’, and were limited to less than 10 classes (Persson et al., 2004; Heikkinen et al., 2010). The emergence of deep learning networks in computer vision, combined with greater data availability, led to a large number of publications combining deep learning with a variety of sensors and data acquisition platforms (Fricker et al., 2019; Kattenborn et al., 2021; Mäyrä et al., 2021; Weinstein et al., 2023). A defining challenge of individual species classification is the fine-grained nature of the task, with subtle differences among co-occurring species, often within the same taxonomic genus. This challenge is compounded by a lack of training data, especially for rarer species, and the natural long-tailed nature of biodiversity leads to a massive imbalance between dominant and rare classes. While there are recent efforts to combat class-imbalance in machine learning for ecology (Nguyen, Demir & Dalponte, 2019; Hemmerling, Pflugmacher & Hostert, 2021; Miao et al., 2021), there remains significant areas for improvement before models can be used for operational analysis at the landscape and continental scales.

Data science competitions are a unique way to advance image processing methods for particular applications (Carpenter, 2011). These competitions provide a standardized dataset and criteria for evaluation, and have the potential to draw expertise from different application domains because of the focus on data science tasks that are found in many applications (Dorr et al., 2016; Marconi et al., 2019; Van Etten, Lindenbaum & Bacastow, 2019). While competitions have allowed for the advancement of many applications in data science, ecology is just beginning to use this format for democratizing method-building (e.g., Humphries et al., 2018; Little et al., 2020), largely due to the recent availability of large, openly available ecological datasets such as from the National Ecological Observatory Network (NEON).

The National Ecological Observatory Network is a 30-year effort of the National Science Foundation to collect standardized organismal, biogeochemical, and remote sensing data over 81 sites in the US from 20 distinct ecoclimatic domains (Schimel et al., 2007). The data provided by NEON covers a broad array of ecosystem components including field data on trees and associated airborne remote sensing imagery. NEON data is ideal for use in data science competitions because it is openly available, well documented, and part of a massive, continental-scale data collection effort. Therefore, methods and lessons learned from the competition can be applied to a large-scale open dataset being used by large numbers of researchers.

The first competition using data from the NEON was run in 2017 and was aimed at generating species predictions of individual tree crowns in a single temperate forest (Marconi et al., 2019). The 2017 competition was instrumental in advancing methods and in establishing a framework for providing data and evaluating submissions. The 2017 competition identified the most effective methods for delineating tree crowns from airborne remote sensing data, aligning delineations to field data, and assigning a species label to delineations. The results of the 2017 competition showed the most successful approaches to species classification require data cleaning to remove noise and outliers prior to analysis, and incorporate uncertainty in predictions show the most promise for future applications.

While the 2017 competition was an important step towards better methods for converting remote sensing to ecological information, it has limited practical application because it was limited to a single site and relied on a small and labor-intensive field dataset collected. Classification methods will be most useful in generating ecological data of individual trees if they achieve high accuracy when: (1) trained on standard forest inventory and remote sensing data, (2) applied across large spatial scales and diverse forest types, and (3) when making predictions in forests where the models have not been trained. The complexity of using standard forest inventory data across multiple sites presents challenges for model performance because of the highly imbalanced multi-species datasets, differences in the species present at different sites, and variability in the remote sensing data due to differences in conditions when the data were collected. While the classification remains critical to the needs of ecologists, an expansion in the diversity of sites and data is required to effectively achieve this task.

To address these needs, a new iteration of the 2017 competition was run that focused on classification using a dataset that allowed for within and cross site evaluation using multiple metrics. This current iteration of the competition uses data from three NEON sites in the southeastern United States to compare how well methods perform on standard forest inventory data at one site, and how well methods perform when applied to a new site. Here we present the results of the competition that includes a comparison of scores from participating teams, a summary of the methods used, and a discussion of how this competition advances our ability to classify individual trees using existing inventory and remote sensing data.

Materials and Methods

Portions of this text were previously published as part of a preprint (https://doi.org/10.1101/2021.08.06.453503) (Graves et al., 2021).

Study sites

The competition used multiple NEON data products from three sites in combination with data collected by members from our research team. The three NEON sites in the southeastern United States (Fig. 1) used in this study are part of three separate NEON ecoclimatic domains and represent distinct environmental, geographic, and vegetative characteristics (Thorpe et al., 2016). The Ordway-Swisher Biological Station in Putnam County, Florida (OSBS, Southeastern domain, 03) is a mixed forest of hardwood and conifers and is primarily managed for maintaining upland pine forests. The canopy of the pine forests is dominated by Longleaf pine (Pinus palustris) and subcanopy Turkey oak (Quercus laevis) with a grass and forb understory. The forests are on deep sandy soil and are managed with prescribed fires at 3–4 year intervals (Krauss, 2018a). More mesic forests are also present at the site, specifically around large water bodies, and contain a mix of pines and hardwood species (see site species list in Appendix A). Mountain Lake Biological Station in the Appalachian mountains of Virginia (MLBS, Appalachians and Cumberland Plateau domain, 07) is a high-elevation forest typical of Southern Appalachia with a closed canopy dominated by Red maple (Quercus rubrum) and White oak (Quercus alba) (Krauss, 2018b). In MLBS, pine species are rarer than at Ordway Swisher Biological Station (OSBS) and Talladega National Forest (TALL), and there is a greater abundance and diversity of canopy hardwood species (see site species list in Appendix A). Talladega National Forest in west-central Alabama (TALL, Ozarks complex domain, 08) is similar to OSBS in management regimes and species in the upland longleaf and loblolly pine forests (Pinus palustris and Pinus taeda, Krauss, 2018c). Similar to OSBS, TALL has deciduous and mixed forest types with a closed canopy and variety of hardwood species in more mesic areas (see site species list in Appendix A). These wetter forests in TALL have some species in common with MLBS (e.g. Liriodendron tulipifera and Quercus alba). In our species dataset, there are 11 species in TALL that are found in either MLBS or OSBS, and 10 species that are only in the TALL dataset.

Figure 1 Map of the three study sites and domains of the National Ecological Observatory Network (NEON).

The sites are part of three separate NEON ecoclimatic domains and represent distinct environmental, geographic, and vegetative characteristics. The map of the USA shows the NEON domains. To evaluate the ability for methods to apply within sites, the Ordway Swisher Biological Station (OSBS) and Mountain Lake Biological Station (MLBS) were used for training and testing (green circles). To evaluate the ability for methods to apply to new sites, the Talladega National Forest (TALL) was only used for testing (orange triangle).

NEON data

The competition used NEON data from two standard collections; remote sensing data and field-collected data. The remote sensing data were generated by the NEON Airborne Observation Platform (AOP) and are provided as four different products, each one measuring different properties of the vegetation and the ground surface (Appendix A, Table A1). The AOP data products are high-resolution orthorectified camera imagery (RGB), discrete return LiDAR point clouds (LAS), LiDAR-derived canopy height model raster (CHM), and spectrometer orthorectified surface directional reflectance—mosaic hyperspectral surface reflectance (HSI). The data products were downloaded using the NEON API and the neonUtilities R package (Lunch et al., 2020). We used the most recent data from the start of the competition: April 2019 for TALL, September 2019 for OSBS, and May 2018 for MLBS. NEON aims to collect airborne data for a minimum of 100 sq km at each site during peak vegetation greenness, when the solar angle is above 40 degrees, and with less than 10% cloud cover (Kampe et al., 2010). For this competition, 20 m × 20 m image subsets were extracted from the original 1 square km tiles downloaded from NEON. Each 20 m × 20 m image subset is either associated with a NEON field plot or with trees that occur outside NEON field plots but were manually mapped in the field by the research team.

The species labels in the training data were collected through the NEON Terrestrial Observation System (TOS). The data contain information on individual tree identifiers, location of trees relative to sampling locations (i.e., distance and azimuth from a central location), species and genus labels, and measures of salient structural attributes. The field attribute that was directly used in the competition was the taxonomic species information that is described by its scientific name, which includes a genus and species classification. To simplify the taxonomic species information, each scientific name is simplified to its unique taxonomic identification code (taxonID). More information about the data products and the field data and the list of species classes and taxonomic codes is provided in Appendix A.

Individual tree crown data

Participants were given bounding boxes labeled with the taxonomic species in the training dataset and generated species predictions for unclassified bounding boxes in the test dataset. Each bounding box represents an individual tree crown (ITC) and was generated by the research team since they are not a standard NEON product. Because ITC data are time-intensive and difficult to generate, the research team used a combination of two different approaches to produce both a reasonably large number of labeled data for training and precise data for evaluation. Both ITC datasets were generated by experts who are familiar with the ecology of the sites.

To generate ITC data for training the research team in the lab visualized multiple remote sensing datasets and field inventory data from NEON for all tree crowns in the 20 m × 20 m image subsets to draw boxes round individual tree crowns. The 20 m × 20 m subsets were located at NEON field plots (specifically over a subplot of the distributed plots) for which surveys of geolocalized tree stems were available from the NEON vegetation structure dataset. The locations of these plots was determined by NEON through a stratified-random spatial balanced design as to capture the variation of the site and allow for robust statistical analysis of the data (Thorpe et al., 2016; Barnett et al., 2019; Meier, Thibault & Barnett, 2023). In the training data, the average distance from one plot to its nearest plot is 246 m. Plot data and locations are available from NEON and maps of each site and the locations of the plots are included in Appendix A.

The training data had 409 ITCs from 39 plots at OSBS and 648 ITCs from 46 plots at MLBS (Fig. 2). Each ITC bounding box was assigned a species label based on the location of individual stem data in the NEON vegetation structure dataset that was determined to match the ITC. To minimize the chance of mislabelling bounding boxes we limited assignment of species labels to boxes that: (1) could clearly be linked to a single stem from the field or a single species in cases where multiple conspecific stems could be candidates; (2) where the field stem was not labeled as fully shaded; (3) where the field stem was not labeled as dead; and (4) where the height of the field stem was not more than 4 m lower than the maximum value of the LiDAR-based canopy height model within the bounding box (when field stem height was available). Those bounding boxes that did not qualify were labeled as unknown species and were not used for the species classification task.

Figure 2 Data used for classification training and testing.

Data listed for each of the three sites from the National Ecological Observatory Network (NEON); OSBS, Ordway Swisher biological station; MLBS, Mountain Lake Biological Station; TALL, Talladega National Forest. Example plots are from TALL. Only RGB data are shown, but RS data include all four remote sensing data products (Appendix A) and are given in the train and test dataset. For each site, the numbers are (1) the number of individual tree crown delineations (ITC), and (2) the number of RS 20 m × 20 m plots (in parentheses). For ease of visualization, the image of the Test-submitted only shows the probability for four taxonID classes for one ITC.

To generate even more precise ITCs for evaluation, the research team created bounding boxes and identified species for a non-random sample of tree crowns directly from the field. These field-based ITCs are not directly overlapping with the NEON plots. The testing data had 218 field-based ITCs at OSBS, 39 at MLBS, and 104 at TALL. More information about how ITC data were generated and how they were related to the field data from NEON is provided in Appendix A.

Importantly, for this study ITC data can be summarized as polygons that represent the spatial extent of individual tree canopies. We provided 2-dimensional rectangular polygons (i.e., bounding boxes) with four vertices at the maximum North/South and East/West directions as a proxy of individual crowns. This strategy is different from what is commonly used for forest and remote sensing methods that use more detailed polygons with many vertices to delineate more precise crown boundaries and shape (e.g., Dalponte et al., 2015). We provide bounding boxes rather than multi-vertex polygons because boxes are a common output of most computer vision methods to identify, extract, and classify objects in an image (Wäldchen & Mäder, 2018). In this way the competition allowed for models to be developed on training data from separate crown detection and delineation methods.

Solicitation and team participation

The competition was announced on February 3rd, 2020 and advertised to individuals and communities focused on remote sensing, image processing, and forest ecology. We also contacted the 109 people who had registered from the 2017 competition. In total, there were 130 registrations for this second competition. Submissions were received from four participating teams (Appendix B, Table B1).

All teams were allowed up to four submissions per task. Submissions made prior to the final submission were evaluated and scores were returned. Pre-submissions were allowed to ensure submissions were properly formatted and provide teams with feedback on model performance. The final submission deadline was extended by 2 months after the train and test data were released. This was done to allow teams more time to work with the data given the challenges associated with COVID-19. The number of pre-submissions was limited to reduce the chance of artificially increasing performance indirectly by iteratively learning method performance from the test set. The number of pre-submissions varied by team, with five pre-submissions from the Fujitsu and Intellisense CAU teams (an additional submission allowed due to timeline extension), four submissions for Jeepers Treepers, and two for Más JALApeñoS.

The original intent of the competition was for individual teams to submit short methods and results articles describing the approaches and performance of their own methods. However, due to the COVID-19 pandemic this became untenable for most teams, with only one team, Jeepers Treepers, submitting the associated companion article (Scholl et al., 2021). For details of Jeepers Treepers methods on the below tasks see Scholl et al. (2021). For details of all other teams’ methods (and summaries of Jeepers Treepers) see Appendix B.

Classification task

The data were split into training and testing datasets where the training data allowed for the development and self-evaluation of models and the testing data was used to evaluate the team methods. Training data included ITCs for the OSBS and MLBS sites, which consisted of 1057 ITC delineations with taxonomic species labels for 85 plots and all remote sensing data products (clips of 20 m × 20 m around each plot; Fig. 2, “Train”). Data were split at the plot-level where all ITCs within a plot were assigned as train or test and therefore spatially distant from each other. This was to reduce the effects of spatial autocorrelation in model development due to the similarity of neighboring pixels (Karasiak et al., 2022). Participants could use any of the remote sensing and field data for training their models since this represents a common scenario where models are developed using data from inventory plots. No TALL data were provided in the training data. The testing data provided to the participants were 353 separate plots with associated remote sensing data, and 585 ITC delineations at the OSBS, MLBS, and TALL sites (Fig. 2, “Test-provided”). The ground truth species labels were withheld from the teams and used for evaluation by the research team (Fig. 2, “Test-ground truth”). Participants submitted the probability of each ITC belonging to one of the taxonomic classes (Fig. 2, “Test-submitted”). The predictions were submitted as a probability from 0 to 100% that the ITC belonged to the associated species class. Providing the ITC bounding boxes kept this task focused on classification methods rather than having participants also incorporate detection and delineation approaches prior to or after classification.

Significant features of this dataset, and forest remote sensing data in general, are class imbalance in the training data, and a difference in species composition and relative abundances between the training data and the test data. Due to the nature of these data, the ability to train on imbalanced data and predict species with species identities and abundances that differ between the training and testing datasets is an important challenge addressed in the competition. The training dataset for the OSBS and MLBS sites had a total of 33 distinct species classes and two genus classes where the species was unknown (Pinus and Quercus), ranging from 1–302 individuals per class (Fig. 3). This distribution represents the composition and relative abundance of canopy trees in the NEON plots and therefore the data available from forest inventory plots that are used to develop and test classification models. The test data for OSBS and MLBS both show unequal distributions of data among species classes. The test data for both sites include 15 species in the training data, and both sites include species in the test data that are not part of the training data (OSBS: 11 species, MLBS: five species). Furthermore, while the test data for TALL has less imbalance across the species classes than the training data at OSBS and MLBS, it includes only 10 of the species from the training data and introduces 11 new species that are not part of the training data (Fig. 3 as the “Other” class). All new species in the test data have few samples and therefore could not be included in the test and train data. In this way, the external TALL site tests not only the ability of the models to be applied to new remote sensing data, but also to a new site with different species composition.

Figure 3 Distribution of samples and reflectance.

(A) Distribution of samples per species class (taxonID) for each dataset. The number of samples for each taxonomic class differs for all sites. Taxonomic class is arranged based on the number of data points in the train data. (B) Hyperspectral reflectance for each data group. Reflectance sampled from 100 random pixels of 10 random 20 m × 20 m plots for each site in the training and test data. Mean (thick lines) and standard deviation (vertical lines) are calculated for each of the three data groups.

Two additional challenges for applying methods to the untrained site were differences in species composition and spectral variation among sites. Species that occur in the test data but not in the train data are grouped together in an “Other” class in the test data. Creating a mixed-species “Other” class that contains species with low samples is a common practice in species mapping because there is insufficient data to accurately train and test each class individually (for example Baldeck et al., 2014). Participants were allowed to include a species class with the label “Other” in their submissions. The “Other” class can be used to indicate a probability that an ITC is a species that is not represented in the training data and is therefore likely a new species in the test dataset that was not seen in the train dataset. Finally, spectral differences among the sites and training and testing data are also an important feature of the dataset that could impact the ability to apply methods to an untrained site (Fig. 3).

Evaluation of the classification task

Classification was evaluated with three class-level metrics to assess the performance across teams; two hard-classification metrics that require a single species class label for each crown (Grandini, Bagli & Visani, 2020) and one soft-classification metric that uses the probability of a crown belonging to any trained species. Accuracy, a common metric in remote sensing classification studies, is the number of samples correctly predicted out of the total number of samples. Since accuracy does not take into account the class-level scores, accuracy is influenced by variable performance and sample size of classes. In this dataset, the class PIPA2 (Pinus palustris) is the dominant class so accuracy is heavily influenced by the model performance for that class. Alternatively, a common classification evaluation metric that is not influenced by class sample size is macro F1. Macro F1 is an aggregate F1 metric of class-level performance, where F1 is the harmonic mean of class precision and recall and is given by Eq. (1), where P is precision, R is recall, TP is the number of true positives, FP is the number of false positives, and FN is the number of false negatives. Macro F1 is the unweighted mean of class F1 and is given by Eq. (2), where C is the set of species classes, Pc is the precision of species class c, Rc is the recall of species class c, and |C| is the number of elements of set C. Macro F1 is a useful evaluation metric when there is imbalance in the class size and variable prediction performance. In this dataset, the class PIPA2 has the same influence as each class in the macro F1 score.

(1) F1=2⋅P⋅RP+R=TPTP+0.5(FP+FN)

(2) macroF1=1|C|∑c∈C⁡F1c=1|C|∑c∈C2PcRcPc+Rc.

The final model-level metric is cross entropy loss, which is a metric commonly used to quantify the performance of multi-class classifiers where all samples are predicted with a probability of belonging to each class. The metric measures the degree of uncertainty in the predictions of the model. Cross entropy loss (Shannon, 1949; Chen, Kar & Ralescu, 2012) is a good measure of model robustness particularly in cases where new classes are introduced into the test set because the metric can capture how the model responds to an increase in test data entropy. Models that have a stronger ability to differentiate between learned classes and new classes have lower cross entropy loss scores and can be considered more robust.

Finally, a confusion matrix of all predictions and class-level precision and recall scores were calculated for all team predictions combined to identify classes that are commonly confused across methods. Precision, or user accuracy, is the percentage of instances classified as positive that are actually positive. High precision means a low commission error for the class where there are few predictions that are not true. Recall, or producer accuracy, is the percentage of positive instances correctly classified as positive. High recall means a low omission error for the class where there are few missing predictions. Evaluation scores and confusion matrices were calculated with the scikit-learn package for python (Pedregosa et al., 2011). The evaluation code is available in the Supplemental Material.

Classification algorithms

A gamut of classification algorithms were used in the competition, with three teams favoring neural network-based approaches and two teams favoring decision tree-based approaches (Table 1). The winning method from the 2017 competition used principal components analysis (PCA) to reduce the dimension of the HSI images to 40 features, and then used an ensemble of a random forest classifier and a gradient boosting classifier (Anderson, 2018). This method was used to generate baseline results.

Table 1 Classification evaluation metrics for participating teams.

Team	Method	Data used	Accuracy	Macro F1	Cross entropy loss	
Fujitsu satellite	Two-stage fully connected neural network	HSI	0.55	0.32	3.6	
Intellisence CAU	1D-convolution neural network	HSI, CHM	0.52	0.24	7.0	
Más JALApeñoS	Extreme gradient boosting	HSI, CHM	0.50	0.14	2.4	
Jeepers Treepers	Two-stage neural network: RetinaNet + multimodal neural network	RGB, HSI, LiDAR point cloud	0.46	0.09	9.2	
Stanford-CCB (baseline)	Random forest and gradient boosting ensemble	HSI	0.44	0.13	7.4	
Note:

Scores are from test data from only the OSBS and MLBS sites. Lower scores are better for cross entropy loss. HSI, hyperspectral reflectance; CHM, canopy height model; RGB, True color image. Bold values indicate the best score among the teams for each evaluation metric.

Only one team used only the HSI data and all other teams use LiDAR data, either as the CHM or the point cloud. The methods are summarized here and additional details are provided in Appendix B. The Fujitsu Satellite team used only the HSI data in a two-step process. First, they used a neural network to encode pixel HSI data in a 2,048-dimension feature vector. The data was clustered to create crown-level feature vectors. The crown level features were put through a 3-layer fully connected neural network with Rectified Linear Unit (ReLU) activation and softmax output layer for the final species classification. The Jeepers Treepers team fused RGB, HSI, and LiDAR data into a neural network model. Their method first used RGB crown data to train a pre-trained (from ImageNet dataset) ResNet convolution neural network (CNN). The vector of probabilities derived from the ResNet was concatenated with the HSI reflectance pseudo-waveform data from the LiDAR point cloud. The concatenated vector was fed through a two-layer multi-layer perceptron with a customized soft-F1 loss function for final classification, and predictions with high uncertainty were labeled as “Other”. The Más JALApeñoS team’s method applied the Extreme Gradient Boosting decision-tree method to HSI data that was first filtered at a pixel level using LiDAR heights. The height-filtered pixels were further filtered using PCA-based outlier removal before application of PCA based dimension reduction. The dimensionally reduced data was run through the Extreme Gradient Boosted model with parameters chosen with a partial grid search. Pixel class probabilities were averaged for the final crown classification. The “Other” class was generated during training by grouping less abundant species into a single “Other” class. Finally, the Intellisense CAU team’s method was based on a one-dimensional CNN applied to HSI pixels. Small classes were resampled to handle the imbalance. The CNN consisted of a convolutional layer, max-pooling layer, a fully connected layer and output. The output was filtered using LiDAR data to remove ground pixels.

Results

Overall performance

For the trained sites (OSBS and MLBS) accuracy of team methods ranged from 0.46–0.55 and was higher than the baseline random forest and gradient boosting ensemble method (accuracy = 0.44, Fig. 4, Table 1). For all teams, macro F1, the metric that equally weighs all species classes, was considerably lower than accuracy (0.09–0.32), and all but one team had higher macro F1 than the baseline (0.13). Cross entropy loss, that takes into account uncertainty in the prediction, ranged from 2.4–9.2, with all but one team outperforming the baseline score of 7.4 (lower is better). The Fujitsu Satellite team’s two-stage fully connected neural network approach had the strongest performance for the two hard-classification evaluation metrics (accuracy = 0.55, macro F1 = 0.32) and the second-best performance for cross entropy loss (cross entropy loss = 3.6, Table 2). The Más JALApeñoS extreme gradient boosting method showed the best performance for cross entropy loss (cross entropy loss = 2.4).

Figure 4 Classification evaluation metrics for all teams at trained and untrained sites.

Higher scores for accuracy and macro F1 and lower scores for cross entropy loss indicate better performance.

Table 2 Prediction metrics for taxonomic class of all team predictions.

TaxonID	Precision	Recall	F1	
ACRU	0.16	0.38	0.23	
CAGL8	0.07	0.01	0.02	
LITU	0.57	0.48	0.52	
Other	0.31	0.22	0.26	
PINUS	0.03	0.08	0.04	
PIPA2	0.66	0.89	0.76	
PITA	0.17	0.01	0.02	
QUAL	0.30	0.20	0.24	
QUGE2	0.13	0.15	0.14	
QULA2	0.35	0.34	0.34	
QUNI	0.18	0.11	0.14	
QURU	0.32	0.50	0.39	
ROPS	0.70	0.30	0.42	
TSCA	0.80	0.16	0.27	
Note:

Values from aggregated confusion matrix. Taxonomic classes with the highest value for each metric are bolded. Taxonomic classes without predictions (with a value of 0) have been removed from the table (ACSA3, FAGR, NYSY, QUERC, QUHE2, QUMO4). Precision is the inverse of commission error and recall is the inverse of omission error.

All methods performed substantially worse on the untrained site (TALL) than the trained sites (OSBS and MLBS), with accuracy ranging from 0.07–0.32 and macro F1 ranging from 0.02–0.18. The highest scores were from the Fujitsu Satellite team’s two-stage fully connected neural network and the lowest accuracy from the Jeppers Treepers’ RetinaNet method. While cross entropy loss scores were better (lower) for all teams on the trained sites, the two methods with the lowest cross entropy scores performed similarly for the trained and untrained sites (Fujitsu Satellite = 4.6 and Más JALApeñoS = 2.8). Since cross entropy loss takes into account the uncertainty in predictions, these results indicate having an uncertain model may be advantageous when applying it to new sites.

Results by species

Model performance varied widely for predictions of individual species classes, with the general pattern of better performance for the most common species and poorer performance for the least common species (Fig. 5). The most common species in the dataset (Fig. 3) is PIPA2 (Pinus palustrus, Longleaf pine), which is a dominant canopy species in the conifer forests in parts of OSBS and TALL. For all team methods, PIPA2 was the best-scoring taxonomic species class, with F1 scores ranging from 0.73–0.86 in the trained sites. For the aggregate predictions for all teams, recall (0.89) for PIPA2 was higher than precision (0.66, Table 2), which indicates that most models tend to over predict PIPA2 relative to other species. As with the overall accuracy metrics, all methods predicted PIPA2 more accurately at the trained sites than at the untrained site (F1 = 0.12–0.54), showing a consistent pattern of a decrease in F1 of approximately 0.35 for all teams. The Jeepers Treeper’s RetinaNet method showed the largest difference between trained and untrained PIPA2 performance (F1 on trained = 0.80 and untrained = 0.12) showing the method was unable to learn features of the species that translated accurately to a new site.

Figure 5 F1 score for species classes and teams.

Taxonomic ID classes are in order of highest to lowest number of samples in the training data with number of samples in parentheses. The “Other” category includes predictions of taxonomic ID classes that were not present in the training data but were present in the testing data, and therefore represent the ability for a method to identify unknown or untrained classes. The shape and the font of the text represents which sites are included in the evaluation (circle & unbolded text = trained sites, OSBS & MLBS; triangle and bolded text = untrained site, TALL). No number is included when the score is zero. Full species information for taxonID labels are in Appendix A.

What contributed to the high macro F1 scores of the top two methods was their ability to predict some of the less abundant species (Fig. 5), specifically Liriodendron tulipifera (LITU, Tulip tree), Tsuga canadensis (TSCA, Eastern hemlock), and Robinia pseudoacacia (ROPS, Black locust). For example, these methods had F1 scores of 0.75–0.77 for LITU in comparison to other teams and the baseline where F1 was 0–0.42. The top methods were responsive to the potentially distinct spectral signature of LITU based on it being taxonomically unique as the only species in the Liriodendron genus, regardless of the low number of samples in the training data (17 train samples, Fig. 3). In addition, TSCA, a conifer species found only at the MLBS site, was not predicted by three methods (baseline, MaS JALApeñoS, and Jeepers Treepers), yet had F1 scores of 0.5 and 0.57 for the Fujitsu Satellite and Intellisence CAU team methods. A similar result was seen for ROPS, a distinct species because it belongs to a legume family, Fabaceae.

All methods performed better than the baseline in predicting a mixed-species “Other” class. Three teams performed similarly in predicting the “Other” species class at the OSBS and MLBS sites with F1 scores of 0.21–0.27 (Fig. 5). The MaS JALApeñoS team, which created an “Other” class in the training dataset by grouping classes with fewer than three samples saw a big difference between the trained and untrained sites (F1 = 0.41 and 0.03, respectively). The Jeepers Treepers team use post-processing by assigning predictions with high uncertainty as “Other”, which resulted in similar scores in the trained and untrained sites (F1 = 0.21 and 0.22, respectively). An encouraging result is that two methods (Fujitsu Satellite and MaS JALApeñoS) had high F1 scores of 0.40 and 0.41, respectively, for the “Other” species class at the TALL site (Fig. 5). These two methods also had the best performance as measured by cross entropy loss, suggesting that the methods that did well when incorporating uncertainty in the prediction are able to identify untrained classes when applied to a new site.

The all-team aggregated confusion matrix (Fig. 6), individual team confusion matrices (Appendix B), and aggregated precision and recall scores (Table 2) show patterns of misclassification within and across the Pinus and Quercus species. For example, commission errors for PIPA2 (precision = 0.66) were mostly due to confusion with a taxonomically and structurally similar Pinus species (PITA) or with Quercus species (QUGE2 and QULA) that co-occurs in the same upland pine-dominated habitat. Confusion within the Quercus genus was also a dominant pattern, as shown by the multiple misclassifications of oak classes in the confusion matrix and precision, recall, and F1 scores generally less than 0.4. There were no correct predictions for three of the oak classes.

Figure 6 Aggregated confusion matrix of all team predictions.

Numbers within the cells and the intensity of color represent the total number of predictions for a given ground truth label. Correct class predictions are along the diagonal with bold text. The background color of the cell corresponds to the true genus category and box outline corresponds to the predicted genus category. Confusion matrices by team are in Appendix B.

Finally, a feature of this dataset was the presence of a Pinus (pines) genus class (PINUS) and the Quercus (Oaks) genus class (QUERC), where the specific species could not be identified in the field. Confusion between the unique pine and oak species (e.g., PIPA2, QULA2) classes and the PINUS and QUERC classes is expected since the individuals with the PINUS and QUERC label are likely one of the species classes in the dataset. Our evaluation did not include any hierarchical structure to account for this feature of the data and no teams chose to include a hierarchical structure in their modeling. Yet, the results show that misclassifications of these genus classes are not limited to their genera, showing that despite having a catch-all pine and oak genus classes, the methods did not learn the taxonomic structure of the data.

Discussion

Evaluation in the scientific literature of many remote sensing approaches to tree species classification tends to focus on a method for a single site and where all species classes are mutually exclusive and known. In this competition, participants were asked to grapple with a challenging classification task, specifically building models using forest inventory training data from multiple sites where there is imbalance in the class sizes, and applying those models to a site where the models have not been trained. By establishing a dataset and evaluation process on which participants with different background can apply their methods, we can evaluate and compare relative performance of different methods for this challenging task. While the high model accuracy scores relative to the baseline winning model from the 2017 competition shows an advancement of methods for tree species classification, many species classes were poorly predicted, especially when applying models to the untrained site.

Two classification methods stood out in how they handled the challenges of imbalance data and application to an untrained site. The first-ranked team based on accuracy and macro F1 scores (Fujitsu Satellite) used a convolution neural network pipeline, consisting of both a pixel and crown-level classifier. Stronger performance of CNNs over shallower machine learning methods for species classification of remote sensing data has been documented in many applications partly due to their ability to learn spatial features and reduced reliance on data pre-processing (Kattenborn et al., 2021). The Fujitsu Satellite team implemented a pixel and crown-level classifier and a unique random spatial data augmentation filter, which is likely key to its success (Appendix A). The first-ranked team based on cross entropy loss (MaS JALApeñoS) used a relatively simple pixel-level decision tree classifier with a partial grid search for best parameters. Despite a more shallow machine learning approach compared to CNNs the Extreme Gradient Boosting method may have been less overfit and while a many species labels may have been incorrect, the certainty of those labels was also low, resulting in a better cross entropy loss score. Finally, while the approaches differed in many ways, both approaches relied on hyperspectral reflectance and reduced the noise and complexity of the data, and extracted relevant features of the 369 hyperspectral bands. This feature engineering was also a key lesson from the original competition in 2017.

An inherent challenge with the classification of ecological data is the imbalance in the data across classes. Most natural forest ecosystems have an unequal abundance distribution of species, with a “hollow curve” shaped distribution where there are a small number of common species, and a large number of relatively rare species (McGill et al., 2007). Ecological datasets often reflect this natural distribution since they are generated by randomly sampling plots in the field. Understanding patterns of taxonomic and functional diversity or evaluating the impact of climate changes and extreme disturbance events on species are examples where poor accuracy of rare species will impact the ability to use the predictions because of the uncertainty in the predictions.

Our results reflect the common outcome of classification on unbalanced datasets, with models generally performing better on classes that have a greater representation in the training data compared to classes with lower representation (Graves et al., 2016; Nguyen, Demir & Dalponte, 2019; Hemmerling, Pflugmacher & Hostert, 2021). Evaluation metrics that are weighted by the number of samples per class, such as accuracy, favor models that are most accurate for abundant classes. However, for many ecological questions and applications, having strong predictions across all species, especially the rare species is important (Leitão et al., 2016; Dee et al., 2019; Cerrejón et al., 2021), and therefore an evaluation score such as macro F1 that equally weighs all species classes, is most appropriate. While model accuracy shows a relatively narrow range in performance among the species, macro F1 shows the distinctly strong performance of two neural network methods (Fujitsu Satellite and Intellisense CAU, Fig. 4) that can discriminate patterns of some less abundant but taxonomically distinct species. In addition, one of the teams that used a network approach and achieved a high macro F1 score (Intellisense CAU) addressed the imbalance by resampling the common classes, which is a common method to reduce the effect of imbalanced data in model training.

Another challenge addressed in this competition was transferability of the model to an untrained site, where the site will most likely contain new taxonomic classes and introduce spectral variation within species, especially if the untrained site is geographically distinct from the training sites. We found that across metrics, all methods performed worse on the untrained site than the trained sites. While the decreased performance is partially due to the change in species classes, even a dominant species (Longleaf pine, PIPA2) was more poorly predicted at the untrained site. This suggests that regardless of differences in species presence and abundance between sites, the spectral and structural signatures of individual species (caused by sensor calibration, atmospheric conditions, seasonal differences, or inherent differences in species foliar and structural properties) are sufficiently different to hinder model performance.

An encouraging result was the presence of methods with significantly lower cross entropy loss scores than other methods (gradient boosting by Más JALApeñoS and a neural network by Fujitsu), and that scores were similarly low for both the trained and untrained sites (Fig. 6). A low cross entropy loss score means that a method was confident with its correct predictions and unconfident with its incorrect predictions. Methods that score low in cross entropy loss could be most useful in transferring to new sites because low confidence in a prediction could indicate the presence of a new species.

The imbalance of data and application of methods to an untrained site means classes will be present that are not in the training data. This challenge is often not directly addressed in species classification tasks in the remote sensing fields, but it is studied in computer science as a type of novel class detection (Din et al., 2021). While this competition design could not fully evaluate the ability to detect untrained species, we did employ the use of mixed-species “Other” class that were present only in the test data, and teams used different approaches to predict this class by grouping trained species with low sample sizes or post-processing based on prediction uncertainty. The F1 scores for the “Other” species class are too low for accurately detecting these new species (~0.2, Fig. 5), yet most methods performed better at this task than our baseline approach from the 2017 competition where an untrained site and species were not part of the task. In addition, methods generally performed better at identifying new species in the trained sites (OSBS and MLBS) than the untrained site (TALL). This dataset and competition can hopefully encourage the remote sensing community to continue to confront this real challenge present in individual tree species mapping.

Finally, this classification task was challenging due to limitations and complexities of the data. The complexities reflect the characteristics of data for real-world applications for which robust methods are needed. One common challenge for ecological applications is that the amount of field data for training and testing is often smaller than the optimal amount to train and robustly evaluate algorithms. We believe the most accurate data for training and evaluating crown delineation and classification models comes from laborious field efforts where individual tree crowns are delineated and species are identified in the field. Datasets like these are small and often limited to specific sites and studies. To overcome limited field data and create a sufficiently large dataset for the competition, we generated a large set of image-delineated crowns to use as training data (see Appendix A). The certainty of these image-delineated ITCs, especially for classification, is less than for the field data because of uncertainty in associating information from field data on individual trees with the remote sensing data. The results showing confusion between two very different species (Fig. 6) suggest that some of the training pixels identified as belonging to one species may in fact belong to another, presenting challenges to classification models. We emphasize that this is an inherent challenge in ecological studies since high-quality data, such as the field-delineated ITCs, will always be limited, and therefore there is a need for methods to account for this source of potential uncertainty. Future efforts should be made to support improved alignment between field and remote sensing datasets (Chadwick et al., 2020). For example, when collecting data in the field, there could be an attribute that specifies if a tree has a position in the canopy and is therefore viewable in remote sensing imagery. Additionally, tree crowns could be digitized in remote sensing data while in the field to avoid any uncertainty and build robust datasets (Graves et al., 2018). Algorithmic approaches may also help address these issues including research in image analysis in classification and detection with label uncertainty (Zou, Gader & Zare, 2019; Du & Zare, 2019), active learning for adding new labels or reviewing existing labels and the inclusion of a self- or semi-supervised step in the learning processes (Weinstein et al., 2019; Kattenborn et al., 2021).

Conclusions

This competition engaged researchers in the data science, remote sensing, and ecology communities to train and apply algorithms for a classification task in the context of individual tree crown species across multiple sites. Participants focused on deep learning approaches, many of which were significantly better at cross-site prediction than the best method from the 2017 competition. By comparing predictions from all teams on the same dataset, we found that the deep learning and more traditional decision tree methods can predict the most common class well, even across sites, but more work is needed in methods that can handle imbalanced data, can predict rare species (i.e., those with lower relative abundances), and are robust to identifying the presence of new species when applied to an untrained site.

Supplemental Information

Supplemental Information 1 Details of ITC and field data generation.

Additional information about the Individual Tree Crown (ITC) and field data provided with the 2020 competition.

Click here for additional data file.

Supplemental Information 2 Team methods and results.

The methods used by participating teams for the classification task. This also includes confusion matrices for the classification methods for each team. See the published articles for full details on the methods for some teams (Table B1).

Click here for additional data file.

Thanks to Morteza Shahriari Nia for helping create the foundation for the 2017 competition.

Additional Information and Declarations

Competing Interests

Author Contributions

Data Availability

Yuzi Kanazawa is employed by Fujitsu Laboratories Ltd. Ethan P. White is an Academic Editor for PeerJ.

Sarah J. Graves conceived and designed the experiments, performed the experiments, analyzed the data, prepared figures and/or tables, authored or reviewed drafts of the article, and approved the final draft.

Sergio Marconi conceived and designed the experiments, performed the experiments, analyzed the data, prepared figures and/or tables, authored or reviewed drafts of the article, and approved the final draft.

Dylan Stewart conceived and designed the experiments, performed the experiments, analyzed the data, prepared figures and/or tables, authored or reviewed drafts of the article, and approved the final draft.

Ira Harmon conceived and designed the experiments, performed the experiments, analyzed the data, prepared figures and/or tables, authored or reviewed drafts of the article, and approved the final draft.

Ben Weinstein conceived and designed the experiments, authored or reviewed drafts of the article, and approved the final draft.

Yuzi Kanazawa performed the experiments, authored or reviewed drafts of the article, and approved the final draft.

Victoria M. Scholl performed the experiments, authored or reviewed drafts of the article, and approved the final draft.

Maxwell B. Joseph performed the experiments, authored or reviewed drafts of the article, and approved the final draft.

Joseph McGlinchy performed the experiments, authored or reviewed drafts of the article, and approved the final draft.

Luke Browne performed the experiments, authored or reviewed drafts of the article, and approved the final draft.

Megan K. Sullivan performed the experiments, authored or reviewed drafts of the article, and approved the final draft.

Sergio Estrada-Villegas performed the experiments, authored or reviewed drafts of the article, and approved the final draft.

Daisy Zhe Wang conceived and designed the experiments, authored or reviewed drafts of the article, and approved the final draft.

Aditya Singh conceived and designed the experiments, authored or reviewed drafts of the article, and approved the final draft.

Stephanie Bohlman conceived and designed the experiments, authored or reviewed drafts of the article, and approved the final draft.

Alina Zare conceived and designed the experiments, authored or reviewed drafts of the article, and approved the final draft.

Ethan P. White conceived and designed the experiments, authored or reviewed drafts of the article, and approved the final draft.

The following information was supplied regarding data availability:

The data is available at GitHub and Zenodo:

- https://github.com/sjgraves/IDTReeS_competition.

- Graves, S., & Marconi, S. (2020). IDTReeS 2020 Competition Data (Version 4) [Data set]. Zenodo. https://doi.org/10.5281/zenodo.3934932.

- https://github.com/weecology/idtrees_competition_evaluation.

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
