# Peer review of "Data science competition for cross-site individual tree species identification from airborne remote sensing data"

_PeerJ, doi:10.7717/peerj.16578_

## Round 0.1 · original submission · Major Revisions

Both reviewers raise a series of major concerns with your manuscript, including:
- A single measure of accuracy to determine a winner is too simplistic.
- It is unclear whether the accuracy among these teams is affected by spatial autocorrelation in the resulting models, which is crucial for their applicability in real-world scenarios.
- The temporal assessment is not included in the competition.

Please thoroughly address these and the other issues raised by both Reviewers, and reframe your manuscript as an appeal to incentivize young students to STEM, instead of claiming that it 'advances our ability to classify individual trees using existing inventory and remote sensing data'. Should you choose to resubmit, note that I will send your revised manuscript back to the Reviewers and will take a final decision based on their assessment.

Reviewer 1 ·

Basic reporting

The manuscript presented here is derived from a competition where teams addressed the complex challenge of species classification or detection using NEON datasets. I appreciate the manuscript as it is based on a competition, but at the same time, I found it unsatisfactory because it reduced this complex challenge to a single output: accuracy. The performance among teams was similar, and the authors did not discuss or incorporate elements associated with the no-free-lunch theorem. To me, it is meaningless if the more accurate team achieved a score of 0.55, while the less accurate team achieved a score of 0.5 but more quickly. Moreover, it is unclear whether the accuracy among these teams is affected by spatial autocorrelation in the resulting models, which is crucial for their applicability in real-world scenarios. Additionally, the authors claim that "this competition advances our ability to classify individual trees using existing inventory and remote sensing data," but it is unclear how they make this assertion without understanding the actual drivers (e.g., absorption features, forest structure, presence of spectral species) that lead to species classification.

Again, I appreciate the manuscript, as I imagine the participants to be teenagers or college students, but the authors should refine their framework beyond the simplistic measure of accuracy to determine a winner.

Specific comments:
Line 51-55: It would be important for readers to know what you mean by performance (e.g., accuracy, F1 score, etc.). Providing an actual value that exemplifies the performance would also be helpful. For instance, when mentioning performance ranges or the highest achieved performance, it would give a better understanding rather than just using subjective terms like "good" or "bad."
Line 141-143: Reproducibility over time is more important for understanding the classification than spatial reproducibility. Perhaps next time, you could explore whether the methods work in different observation periods.
Line 388: The accuracies are very low, which explains why you did not mention them in the abstract. Maybe you could provide the highest accuracy achieved there, such as 0.55?
Line 541: After skimming the text twice, it is still unclear to me whether the classification was done using only the hyperspectral observations or all available data (RGB, LiDAR, and hyperspectral). It is clear that you provided these data to the teams, but it is not clear if the teams actually used them.
Figure 3: The water absorption bands seem to have been removed, which is acceptable. However, it affects the continuity of the x-axis, resulting in labeling errors. It appears that you need to recreate this figure to correct the issue.

Experimental design

It is not clear how the spatial auto-correlation of the resulting models affect the performance and so the declaration of the winner.

Validity of the findings

Again it is not clear how this competition truly advances our ability to classify species without the understanding of the actual drivers that lead to species classification. As authors, we should avoid the black-box of accuracy in our results, and thus, the authors of this manuscript should explore that.

Additional comments

The temporal assessment is not included in the competition. Are the models traceable over time?

Reviewer 2 ·

Basic reporting

Review on “Data science competition for cross-site individual tree species identification from airborne remote sensing data”

This work introduced a data science competition to help identify effective methods for the task of classification of individual crowns to species identity. The results of four teams taking party in the competition were presented and analyzed in this work. The most challenged part of this competition is that the dataset was highly imbalanced for different species and provides very limited number samples for some species. The contribution of this work was not clear, and there are some points need to be clarified. Consequently, I think the manuscript isn’t ready for publishment.
1. Line 58, “This work shows that data science competitions are useful to compare different methods through the use of a standardized dataset and evaluation criteria, which highlights promising approaches and common challenges, and advances the ecological and remote sensing fields”. This sentence seems to be a common result. This is not a contribution of this work. Maybe the dataset or the evaluation criteria can be the contribution of this work.
2. Line 82-83: Spectral features should be mentioned here.
3. Line 88-89: Some works using CNN models published in the recent three years should be cited here.

4. Line 124, what do you mean by multi-part?
5. Line 130-134, “Classification methods…..” . This sentence is hard to follow.
6. Line 206: The training data are bounding boxes labeled with the taxonomic species. However, normally, individual tree crowns were delineated and then labelled for training. This is a major defect of the dataset.
7. Line 247-248, ‘We provide bounding boxes rather than multi-vertex polygons because boxes are a common output of most computer vision methods to identify, extract, and classify objects in an image’. It’s true. However, precisely tree crown boundaries are necessary to label tree species. The bounding boxes of these boundaries may be used in training samples, and the boundaries may be used in mapping the classification results.
8. The introduction of the classification algorithms was too simple. It's difficult to understand the advantages of the methods, because different images were employed by the four teams and the baseline. Did they used any data augmentation approach to deal with the limited number of tree samples?
9. Line 334: The equations of macro F1 score and F1 score should be provided to help the comparison of accuracies for species.
10. Line 394: the Fujitsu Satellite team’s two-stage fully connected neural network yielded the best performance. The team used only the HSI to deal with the classification of more than 20 species. What contributed to the results? Just the two-stage fully connected neural network? What’s the advantage of this CNN model over other models? It is important to clarify this point.
11. Line 438: Why does the other category represent the untrained species classes? I can't agree with you once the other category includes two or more taxonomic ID classes. In this work, the F1-score of the other category was taken as the index measuring the ability of a method to identify unknown or untrained classes. Meanwhile, the relevant analysis results are not reliable.
12. Line 559-660: This sentence is hard to follow. What does the word 'indicate' mean?
13. Line 577, ‘standardized dataset’. The dataset used in this work is unbalanced between different species, and therefore, is not a standardized dataset. Moreover, the sample number of some species in this dataset is very limited.

Experimental design

No comment.

Validity of the findings

No comment.

---

## Round 0.2 · accepted · Accept

Thank you for the thorough revision of the manuscript and your patience in the entire review process. I have been asked to take over as the prior Academic Editor was unavailable. I hereby certify that you have adequately taken into account all of the reviewers' comments, as I have checked by my own assessment of your revised manuscript. Based on my assessment as an Academic Editor, your manuscript is now ready for publication.

Reviewer 1 ·

Basic reporting

no comment

Experimental design

no comment

Validity of the findings

no comment

Additional comments

I consider that the authors addressed my comments satisfactorily. I still consider that knowing the drivers of why a good classification can be archived beyond a method is more important than a single performance value. However, given the this manuscript emerges from a competition, I understand that the drivers are difficult to evaluate.